# Scalable cryopreservation of infectious *Cryptosporidium hominis* oocysts by vitrification

**Justyna J. Jaskiewicz**[1,2�], **Denise Ann E. Dayao**[2�], **Donald Girouard**[2], **Derin Sevenler**[1], **Giovanni Widmer**[2], **Mehmet Toner**[1], **Saul Tzipori**[2]*, **Rebecca D. Sandlin**[1]*

1 BioMEMS Resource Center, Center for Engineering in Medicine and Surgery, Department of Surgery, Massachusetts General Hospital, Harvard Medical School, and Shriners Children's Boston, Boston, Massachusetts, United States of America, 2 Department of Infectious Disease and Global Health, Cummings School of Veterinary Medicine, Tufts University, North Grafton, Massachusetts, United States of America

º These authors contributed equally to this work.
* Saul.Tzipori@tufts.edu; RDSANDLIN@mgh.harvard.edu

**Data Availability Statement:** The datasets generated during and/or analyzed during the current study are available in the Supporting Information Data files.

## Abstract

*Cryptosporidium hominis* is a serious cause of childhood diarrhea in developing countries. The development of therapeutics is impeded by major technical roadblocks including lack of cryopreservation and simple culturing methods. This impacts the availability of optimized/standardized singular sources of infectious parasite oocysts for research and human challenge studies. The human *C. hominis* TU502 isolate is currently propagated in gnotobiotic piglets in only one laboratory, which limits access to oocysts. Streamlined cryopreservation could enable creation of a biobank to serve as an oocyst source for research and distribution to other investigators requiring *C. hominis*. Here, we report cryopreservation of *C. hominis* TU502 oocysts by vitrification using specially designed specimen containers scaled to 100 µL volume. Thawed oocysts exhibit ~70% viability with robust excystation and 100% infection rate in gnotobiotic piglets. The availability of optimized/standardized sources of oocysts may streamline drug and vaccine evaluation by enabling wider access to biological specimens.

## Author summary

We report here on the first successful method of cryopreservation of *Cryptosporidium hominis* parasites, one of the most common infectious causes of pediatric diarrhea in the developing world. This is significant as lack of cryopreservation methods for this species of parasite hinders progress of scientific and clinical research. In the absence of these methods, a reference isolate of *C. hominis* must be maintained periodically by passage in gnotobiotic piglets. Cryopreservation of this unique isolate is therefore a priority to increase sharing and availability for research, and to prevent catastrophic loss. Cryopreservation is also necessary to facilitate human challenge studies and clinical trials for the evaluation of therapeutics. Here, we successfully cryopreserved the parasite by vitrification using rapid cooling rates achieved in specially designed specimen containers. We showed

**Funding:** The research was funded by the National Institutes of Health award R21AI154026 granted to R.D.S. and S.T., the National Science Foundation award EEC 1941543 granted to M.T. and the Good Ventures Foundation award granted to S.T. Support from R21AI154026 was received by J.J., R.S, M.T., G.W., and S.T., from EEC 1941543 by J. J., D.S., M.T. and R.S., and from Good Ventures Foundation award by S.T. and D.D. The funders had no role in study design, data collection and analysis, decision to publish, or preparation of the manuscript.

**Competing interests:** We have read the journal's policy and the authors of this manuscript have the following competing interests. Patent protection has been filed for the vitrification cassette technology.

that the thawed parasite retained viability and infectivity in gnotobiotic piglets. Notably these methods allowed us to establish a biobank of *C. hominis* oocysts of reference isolate TU502. The availability of biobanked optimized/standardized oocyst preparations may facilitate drug and vaccines evaluation.

## Introduction

Cryptosporidiosis is a globally distributed gastrointestinal illness caused by infection with the apicomplexan parasite, *Cryptosporidium*. In developing regions of the world, *Cryptosporidium* ranks the 4th most common cause of infectious diarrhea of children leading to 8 million cases and 200,000 deaths annually [1–4]. The annual burden from pediatric cryptosporidiosis was estimated at >12 million disability-adjusted life-years accounting for malnutrition [5]. Cryptosporidiosis also causes life-threatening diarrhea among immunocompromised individuals, affecting the HIV/AIDS population disproportionately [6,7]. Outbreaks of cryptosporidiosis in developed nations are on the rise in the last decade [8–11]. Human infections are most frequently caused by *C. hominis* and *C. parvum* [3,12,13], with *C. hominis* accounting for nearly 95% of *Cryptosporidium* infections in some communities [14–16]. In contrast to zoonotic *C. parvum*, *C. hominis* circulates predominantly between humans. Highly infectious and easily transmissible *C. hominis* leads to large outbreaks in association with recreational waters and direct contact with feces [17–19]. No drugs or vaccines exist to relieve the disease burden in the most affected populations. Their development is hampered by many technical limitations, such as lacking methods for cryopreservation and simple *in vitro* culturing platforms, thereby requiring maintenance of laboratory isolates of *C. hominis* by passage in animals. Although successful infections have been established previously in rodent models [20] [21], laboratory maintenance of *C. hominis* is practical only in a gnotobiotic piglet model [22]. However, the cost, complexity and requirement for veterinary facilities and expertise restrict the common use of the gnotobiotic piglet model. Consequently, to our knowledge, only one laboratory isolate of *C. hominis* is currently maintained worldwide. Specifically, the *C. hominis* TU502 isolate derived from a Ugandan patient has been maintained for last two decades by serial passage in gnotobiotic piglets every 2–4 months in the Tzipori lab at Tufts University. The effort invested in propagation of this unique laboratory isolate underscores the need for cryopreservation. *C. hominis* TU502 has been extensively characterized, including its genome sequence [23,24], evaluation of host immune response to infection [20,25–27], susceptibility to therapeutics [28–30] and infectivity to humans [27]. This extensive information justifies designating this isolate the *C. hominis* reference. Cryopreservation of this unique isolate is therefore a priority, to increase sharing and availability, and protection from catastrophic loss. Further, cryopreservation would eliminate batch-to-batch variability among specimens utilized for clinical and therapeutic applications, which currently demands optimization, standardization, and *in vitro* validation for each propagated batch prior to evaluation at testing sites. A controlled infection of human subjects with standardized parasite preparations of known species can ascertain uniformity of the challenge dose and thereby facilitate execution of phase I and II clinical trials.

No effort to cryopreserve *C. hominis* has been reported to our knowledge. The repeated failures to preserve *C. parvum* oocysts by slow cooling [31,32], snap freezing [33–35], or desiccation [33,34] have perhaps dampened enthusiasm for evaluation of these methods in *C. hominis*, given its limited availability. Several methods have been reported for preservation of *C. parvum* but have not been adopted since their publication, whether due to poor stability in storage above -20°C or lack of replication success [35–38]. Prior preservation failures by

classical methods of cooling led us to consider vitrification, an ice-free approach to cryopreservation where specimens are cooled rapidly through the glass transition to form an amorphous solid avoiding the formation of crystalline ice. Previously, we demonstrated successful preservation of infectious *C. parvum* oocysts by vitrification. This was first accomplished using silica microcapillaries (2 μL) and later scaled to larger volumes using specially designed high-aspect ratio cassettes [39,40], which allowed for sample volumes of ~200 μL. In these prior methods, the intracellular uptake of cryoprotective agents (CPA) was enabled by bleaching and additionally heightened by dehydration in trehalose. Although the concentration of CPA accumulated inside oocysts is unknown, cryopreservation of infectious *C. parvum* was achieved in microcapillaries and cassettes in extracellular CPA cocktails consisting of 0.5 M trehalose / 30% DMSO and 0.8 M trehalose / 50% DMSO, respectively. Here, we set to cryopreserve *C. hominis* by rapid cooling technologies, focusing our effort on vitrification in newly optimized cassettes. Due to *C. hominis* resistance to chemical permeabilization, the protocol developed for *C. parvum* is however not compatible. We have overcome this issue here by developing a thermal permeabilization technique which enables uptake of CPA at temperatures >30°C, followed by optimization of a CPA cocktail solution consisting of trehalose and DMSO, and rapid cooling to -196°C. For validation of the developed protocols, *in vivo* infectivity of cryopreserved *C. hominis* was tested in gnotobiotic piglets.

## Results

### Permeabilization of *C. hominis* oocysts to water and cryoprotective agents

The replacement of intracellular water with CPAs is required for optimal cryopreservation, particularly in the case of vitrification, where high water concentrations may lead to lethal ice crystallization upon rapid cooling. The impermeable nature of the oocyst wall prevents CPA uptake and removal of water and is therefore a major obstacle to cryopreservation. Here, to remove excess water from oocysts, the non-permeating CPA, trehalose, was examined. *C. hominis* oocysts responded to hyperosmotic gradient of trehalose by exclusion of intracellular water resulting in oocyst shrinkage, as observed previously with *C. parvum* [39]. Oocyst dehydration progressed as a function of trehalose concentration, leading to 80% of cellular volume loss in a 1 M solution (Fig 1A).

While oocyst permeabilization to CPAs is possible with bleach treatment among *C. parvum* oocysts [39,40], this approach was not effective for *C. hominis*. Specifically, no lethality was observed among *C. hominis* oocysts exposed to toxic concentration of DMSO for 30 min following treatment with varying concentrations of sodium hypochlorite, suggesting that the treatment did not permeabilize oocysts (S1A Fig). As an alternative approach, thermal permeabilization was explored in an effort to melt lipid components of the oocyst wall [41]. Oocysts were incubated with 50% DMSO at either 30°C or 37°C for 30 min. The rapid onset of toxicity suggests CPA uptake occurs under these conditions (Fig 1B). While gradual onset of mortality in the presence of DMSO is observed at 30°C, the effect accelerates sharply at 37°C, leading to measurable toxicity within 5 min of incubation. In contrast, only negligible toxicity is noted at 21°C, suggesting little or no uptake of DMSO. Uptake of CPA at 37°C was also confirmed by microscopic observation of a shrink-swell response, where oocysts show an osmotic response consistent with intracellular uptake of DMSO (Fig 1C). We verified the method of thermal permeabilization to be replicable across different batches of oocysts. In contrast, chemical permeabilization by bleach or alkane treatment produced variable response across oocyst batches and thus was not considered for the development of CPA exposure protocols (S1B Fig).

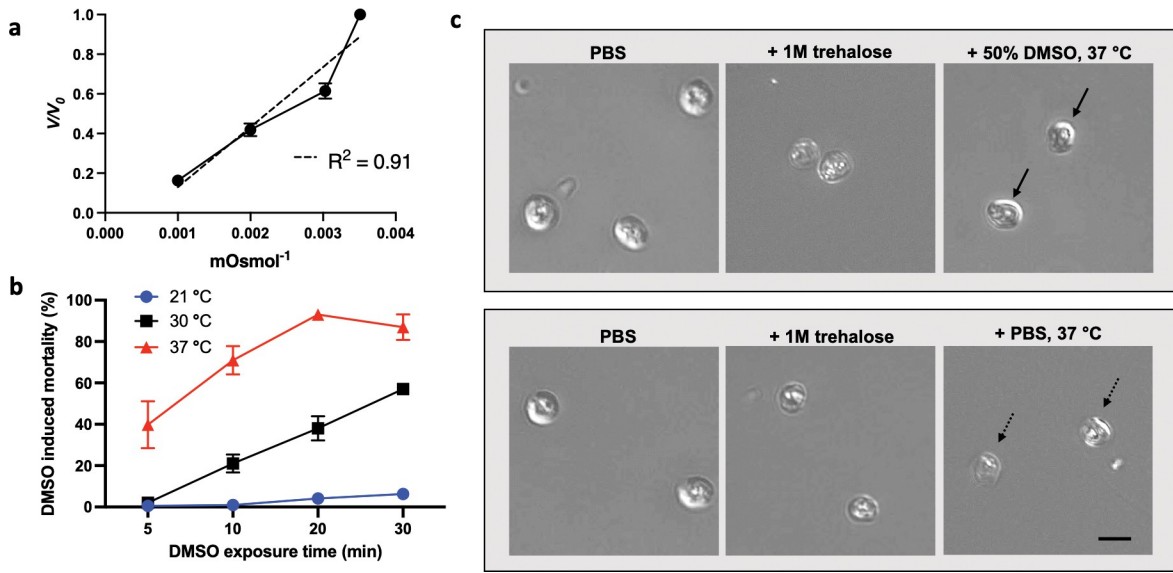

**Fig 1. Permeabilization of *C. hominis* to water and cryoprotective agent. a)** Exclusion of water from oocysts was achieved under pressure of a hyerosmotic gradient of trehalose and is reported as change in oocyst volume ($V$, estimated oocyst volume; $V_0$, starting volume of control oocysts in PBS). The concentration of trehalose (0 M, 0.33 M, 0.5 M and 1 M) is expressed as inverse osmolarity on the X-axis. Dehydration observed in response to the trehalose gradient is substantial (One-way ANOVA; p < 0.0001, F = 200, df = 11), and linear (Pearson's correlation coefficient; $R^2$ = 0.91). Data met requirements of normality (Shapiro-Wilk test; $p$ > 0.21 for all trehalose concentrations) and homoscedasticity (Brown–Forsythe test; $p$ = 0.47) for inclusion in ANOVA analysis. Values indicate mean and error bars indicate standard error (n = 3). **b)** The kinetics of DMSO toxicity were measured with 2-week-old oocysts in response to time and temperature of DMSO exposure. Oocysts were first dehydrated with 1 M trehalose for 10 min and then treated with a solution of DMSO to achieve a final concentration of 0.5 M trehalose/ 50% DMSO. DMSO-induced mortality was measured by PI inclusion using flow cytometry and was normalized to control oocysts treated with trehalose and PBS in lieu of DMSO. Increased temperature significantly accelerates mortality (Two-way ANOVA; $p$ <0.0001, F = 214, df = 2 with requirements of normality homoscedasticity met: Shapiro-Wilk test; $p$ >0.054 and Brown–Forsythe test; $p$ >0.53), which is consistent with increased permeabilization and intracellular uptake of DMSO. Values indicate mean and error bars indicate standard error (n = 3). **c)** DIC micrographs in upper panel demonstrate shrink-swell response of oocysts during the multi-stage CPA loading procedure. Oocysts first shrink in response to 1 M trehalose and swell after exposure to DMSO for 2 min at 37°C (solid arrows). Control oocysts in the lower panel exposed to PBS in lieu of DMSO swell only minimally (dashed arrows) due to dilution of trehalose to 0.5 M and thus reduction of osmotic pressure. Scale indicates 5 μm.

## Development of cryoprotective agent treatment parameters

While our data suggests that intracellular accumulation of DMSO is achieved in thermally permeabilized *C. hominis* oocysts, the concentration of CPA inside oocysts is unknown. This is critical as rapid cooling will likely cause lethal intracellular ice crystallization if insufficient CPA uptake occurs. To maximize intracellular CPA concentration, we examined the toxic effects of a 50% DMSO solution to identify conditions leading to minimal mortality. Based on the observed correlation between *C. hominis* oocyst age and permeability to CPA, where older oocysts respond to permeabilization by bleach (S1B Fig), we sought to evaluate this association in relation to thermal permeabilization. As expected, the observed variability in permeability to 50% DMSO at 30°C is age-dependent ($p$ = 0.006, Fig 2A). However, increasing the permeabilization temperature to 37°C was found to effectively eliminate the age effect of permeabilization ($p$ = 0.13, Fig 2B). Given that permeabilization at 37°C also accelerates DMSO uptake and leads to rapid onset of mortality, a shorter exposure was prioritized. We found that treatment with 1M trehalose followed by a 2 min exposure to 50% DMSO at 37°C reduces variability in mortality across all ages of oocysts (Fig 2B) and does not negatively impact excystation, i.e., the release of sporozoites in the presence of taurocholic acid, a type of bile salt (Fig 2C and 2D). Surprisingly, improved excystation in response to DMSO treatment alone was observed in the presence of taurocholic acid (Fig 2C). Since DMSO is a known solvent for hydrocarbons, we

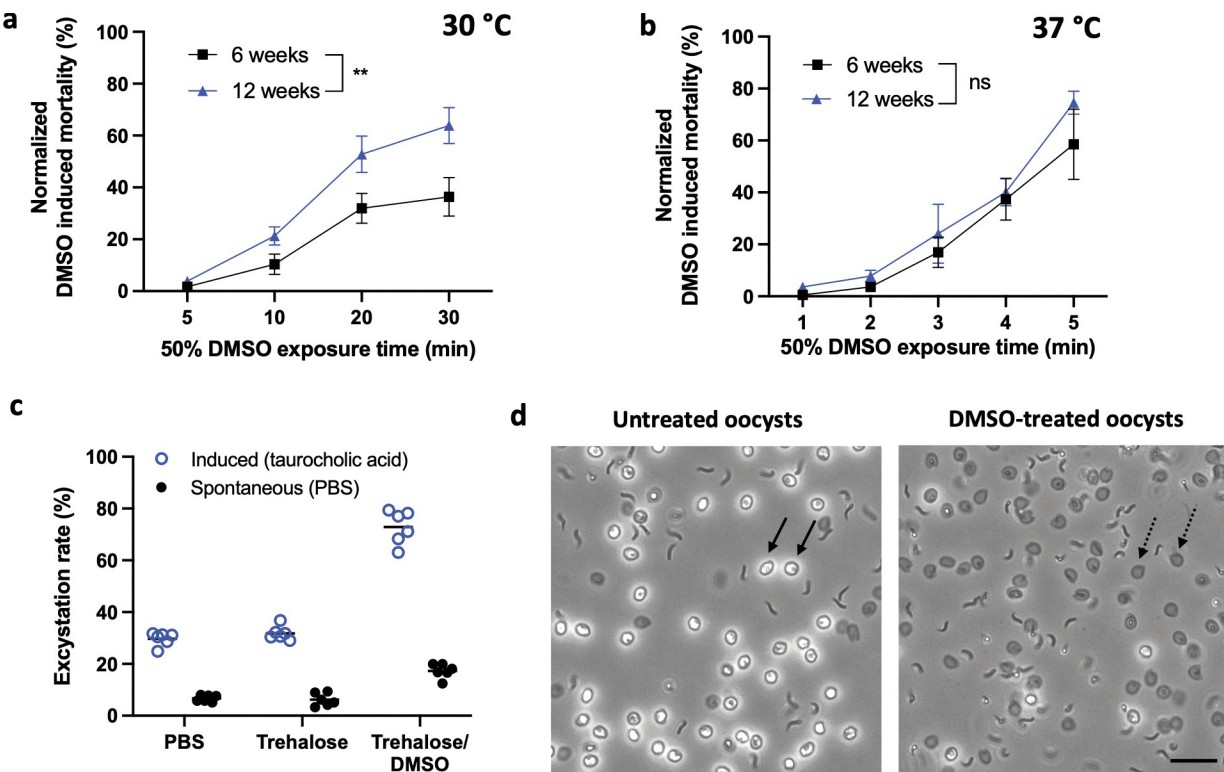

**Fig 2. Optimization of thermal permeabilization and cryoprotective agent exposure.** Matched 6- and 12-week-old *C. hominis* oocysts were dehydrated in 1 M trehalose (10 min) and then exposed to 50% DMSO at **a)** 30˚C for 5–30 min or **b)** 37˚C for 1–5 min. Oocyst mortality was measured by flow cytometry based on PI inclusion and was normalized to control oocysts treated with PBS in lieu of DMSO. A different response between oocyst ages is observed (Two-way ANOVA; **$p = 0.006$, F = 9.8, df = 1, Shapiro-Wilk normality test; $p > 0.52$, and Brown–Forsythe homoscedasticity test; $p > 0.29$). In contrast, exposure to DMSO at 37˚C leads to rapid increase in toxicity with no differential age response (Two-way ANOVA; $p = 0.13$, F = 2.4, df = 1, Shapiro-Wilk normality test; $p > 0.47$, and Brown–Forsythe homoscedasticity test; $p > 0.47$ for all age and time conditions tested). Values indicate mean and error bars indicate standard error (n = 3). ns = non-significant. **c)** After exposure to 1M trehalose (10 min) and 50% DMSO (2 min, 37˚C) or trehalose alone, functional viability was evaluated microscopically using an excystation assay. Excystation rate was calculated as a percent of oocysts which released sporozoites after 30 min incubation with 0.75% taurocholic acid at 37˚C (induced) or in PBS at 37˚C (spontaneous). 'PBS' control incubated under identical conditions in PBS in lieu of CPA determines baseline excystation. Exposure to DMSO substantially increases induced excystation with minimal effect on spontaneous breakdown. Lines indicate mean (n = 6). **d)** Micrographs demonstrate the extent of induced excystation in DMSO-treated oocysts in comparison to untreated oocysts. Solid arrows pointing to refractory oocysts indicate full unexcysted oocysts and dashed arrows indicate empty shells of excysted oocysts. Sporozoites released after DMSO-treatment are of comparable quality to those yielded by untreated oocysts. Scale indicates 20 μm.

suspect it may remove the layer of lipid hydrocarbon reported to rest directly on the oocyst suture [41], thus exposing it more readily to bile acid. Similarly, a previously observed increase in excystation rate after bleaching is likely attributed to removal of fatty acids from oocyst wall [39,42]. The CPA loading protocol consisting of dehydration in 1 M trehalose followed by 2 min incubation in 50% DMSO at 37˚C was therefore selected as the least toxic and most broadly applicable approach for CPA loading across oocyst age groups and batches.

## *C. hominis* oocysts are amenable to vitrification by ultra-rapid cooling

The feasibility of vitrification as an approach to cryopreservation of *C. hominis* was first evaluated in silica microcapillaries (200 μm internal diameter, Fig 3A) to serve as proof of concept that oocysts are amenable to vitrification. Microcapillaries enable exceptionally rapid cooling rates upon submersion in liquid nitrogen, thus enabling vitrification in the presence of relatively low concentrations of CPAs [43]. Indeed, prior work with *C. parvum* demonstrated that

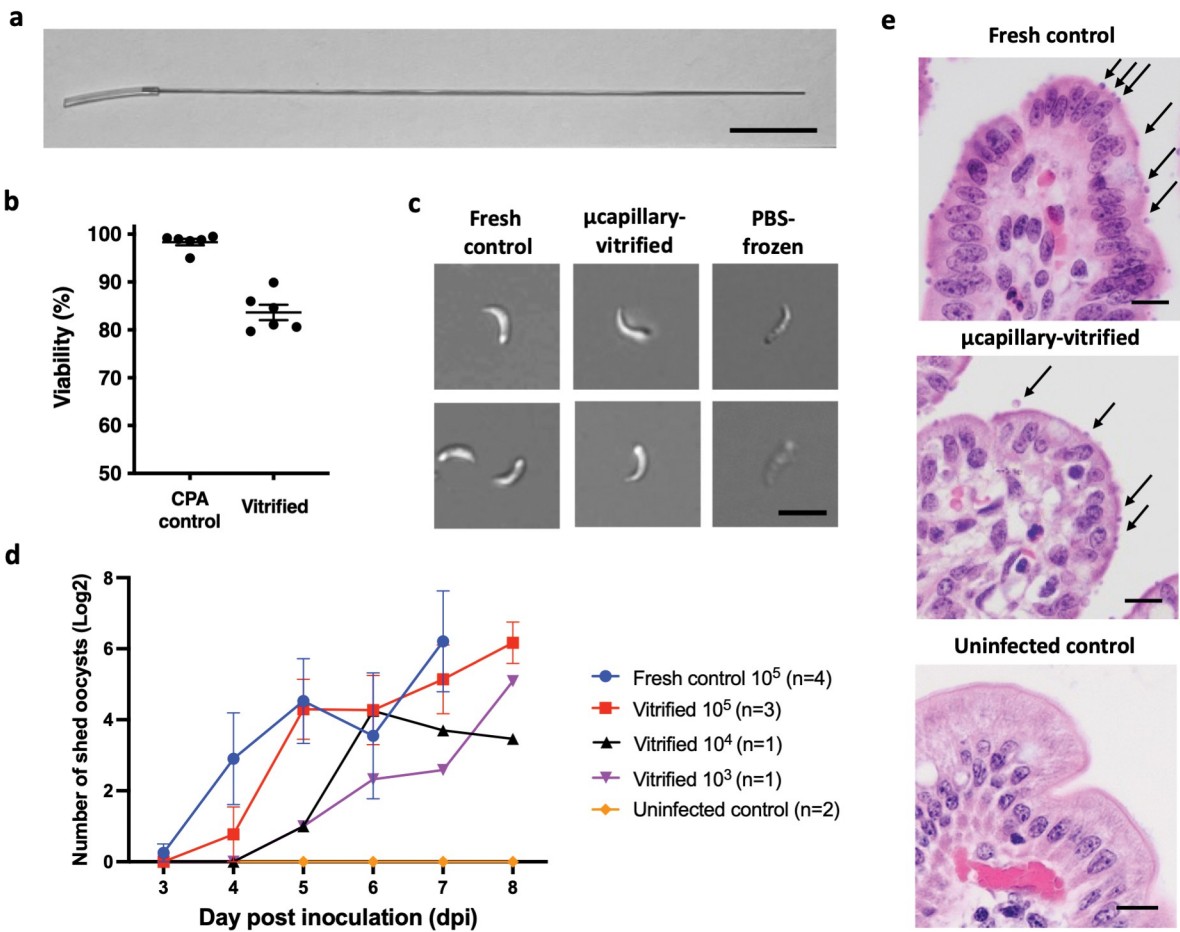

**Fig 3. Cryopreservation of infectious *C. hominis* oocysts by ultra-rapid vitrification. a)** Silica microcapillary used for vitrification. Scale indicates 1 cm. **b)** Viability was quantified using microscopy by means of propidium iodide (PI) exclusion in oocysts exposed to 0.5 M trehalose/ 30% DMSO for 5 min, both before (CPA control) and after cryogenic storage (vitrified). Values indicate mean and error bars indicate standard error (n = 6). **c)** Sporozoites excysted from oocysts vitrified in capillaries remain morphologically similar to fresh controls. Functional viability was assessed based on sporozoite structure, shape and observed motility in reference to fresh and killed control oocysts. Scale indicates 5 μm. **d)** *C. hominis* oocysts are infectious to gnotobiotic piglets after 16–40 months of cryogenic storage. To determine minimum infectious dose, piglets were inoculated orally with either $10^5$ (n = 3), $10^4$ (n = 1) or $10^3$ thawed PI⁻ oocysts (n = 1) in the presence of controls infected with $10^5$ fresh oocysts (n = 4) and an uninfected control (n = 2). Aside from two inocula dosed at $10^5$ oocysts, which were recovered after 40 months of cryogenic storage, the remainder of inocula were stored for 16 months. Fecal shedding of oocysts was determined daily by microscopic enumeration in 30 fields of acid-fast stained fecal smears examined at 1000x magnification. Values indicate mean of log transformed oocyst counts and bars indicate standard error. Untransformed infectivity data for each individual piglet can be found in S2 Fig. **e)** Micrographs of hematoxylin and eosin-stained ileal sections from piglets inoculated with oocysts, either fresh or cryopreserved in microcapillaries for 40 months, and from an uninfected control. Arrows indicate intracellular parasite stages located at the apex of enterocytes. Scale indicates 20μm.

cooling in microcapillaries enables vitrification in the absence of exogenous intracellular CPAs [39]. To determine the feasibility of *C. hominis* vitrification, oocysts were dehydrated in trehalose, followed by addition of DMSO to achieve a final CPA cocktail composition of 0.5 M trehalose/ 30% DMSO. Samples were then cooled at a rate of 250,000°C /min by rapidly submerging the microcapillaries in liquid nitrogen [43]. Oocyst viability among CPA toxicity controls was 98.1 ± 1.5% as evaluated by the propidium iodide (PI) exclusion assay, suggesting a lack of DMSO uptake (Fig 3B). Despite the apparent absence of intracellular CPA during cooling, 84.1 ± 3.7% oocysts were PI negative and presumed to be viable after thawing. Sporozoites excysted from vitrified oocysts were morphologically fit and often motile, resembling

those excysted from fresh control oocysts (Fig 3C). Importantly, gnotobiotic piglets inoculated with oocysts thawed after 16–40 months of cryogenic storage developed a patent infection evidenced by fecal shedding of oocysts (Figs 3D and S2). Evaluation of thawed parasite infectivity at doses varying from $10^3$ to $10^5$ PI$^-$ oocysts per animal indicated that the minimal tested dose obtained from a single microcapillary produced a robust infection. Notably, fecal oocyst shedding started on the same day or shortly after as those infected with a matched dose of fresh control oocysts, indicating that fresh and cryopreserved inocula were of comparable infectivity. Micrographs of ileal sections demonstrating heavy infection of intestinal epithelium are evidence of infectivity after 40 months of cryogenic storage (Fig 3E), which is the longest time tested to date. Verification of oocyst identity was made after recovery from long-term cryogenic storage and subsequent animal passage, by 18S rRNA sequence analysis (S3 Fig). Details of all infections of gnotobiotic piglets with microcapillary-vitrified oocysts originating from five different batches are included in S1 Table. A detailed step-by-step protocol for ultra-rapid vitrification using the microcapillary can be found elsewhere [39].

## Design and fabrication of high-aspect ratio ready-to-use container for vitrification of 100 μL samples

Restriction of the microcapillary volume to 2 μL limits implementation into a clinical and preclinical setting as it prevents storage of multiple inocula in a single device. To overcome the small volume of microcapillaries, a previously described vitrification cassette was used as a starting point for optimization [40]. The concept of the cassette is to utilize a high-aspect ratio geometry to reduce thermal mass and increase cooling rates to those consistent with vitrification. Here, we have redesigned major features of the device to improve its integrity, safety, and user experience (Fig 4A and 4B, with detailed dimensions described in S4 Fig). To first improve device integrity in liquid nitrogen, an acrylic-based adhesive resistant to cryogenic temperatures was used to connect two polycarbonate sheets. When cooled in liquid nitrogen, the device enables transition of a minimum 30% DMSO solution to a vitreous state (Fig 4C) where the estimated cooling rate of $10^{4}$°C /min was extrapolated from published reports [44,45]. New features were also introduced to prevent sample leakage during cryostorage and thawing. Loading ports were outfitted with silicone tabs, both to act as closures, and to enable sample loading with a syringe (S5A Fig). For retrieval of the sample, exit ports were designed as cuttable channels located in the corners of the device (S5B Fig). Their dead-end design blocks entry of biospecimen and therefore prevents its aerosolization during sample retrieval, which improves safety for work with infectious agents. Lastly, the shape of the cassette was redesigned to facilitate evacuation of the sample by centrifugation in a conical tube (S5C Fig). This unloading strategy allows for recovery of 95.4 ± 2.3% of sample volume. Following the changes in dimensions, the cassette volume was reduced to 100 μL however the overall recovery of sample increased by 57.7 ± 6.25% in comparison to the previous design.

## Cryopreservation of permeabilized *C. hominis* oocysts in vitrification cassettes

The use of vitrification cassettes to cryopreserve *C. hominis* oocysts was next examined. Here, four batches of oocysts were examined over the course of 14 weeks using the thermal permeabilization approach. Fig 5A describes the workflow of procedures used in these cryopreservation experiments (for detailed protocol, see S1 Protocol). First, oocysts were dehydrated in 1 M trehalose for 10 min and then suspended in DMSO to achieve a final concentration of 0.5 M trehalose / 50% DMSO. Exposure to DMSO occurred at either 30°C or 37°C for 2-, 5- or 10 min. Rapid cooling was achieved by loading oocysts into a vitrification cassette followed by its

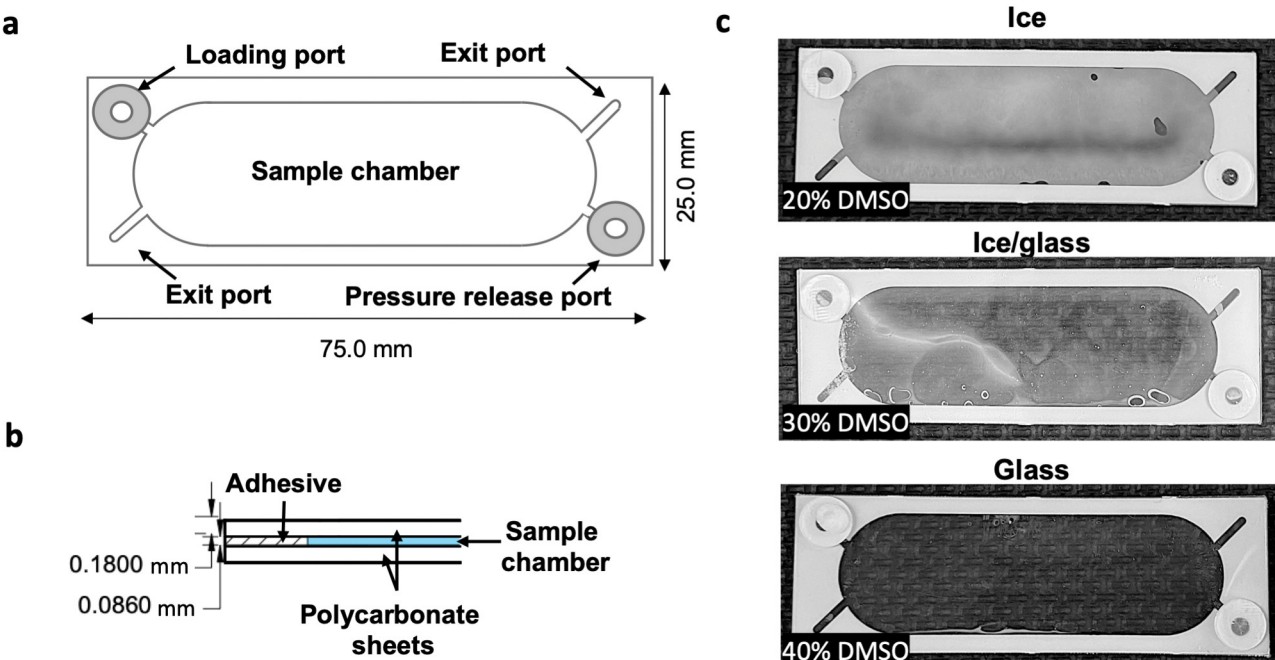

**Fig 4. High aspect ratio cassette enables vitrification of scalable volumes. a)** The outer cassette dimensions are 25 mm by 75 mm with internal volume of approximately 100 μL. More details on dimensions are provided in S3 Fig. Silicone tabs are utilized as self-healing loading ports and pressure release ports, while unloading is performed by cutting exit ports with sterile scissors. Details of the sample loading and unloading technique are provided in S4 Fig. **b)** Cross sectional view of the cassette shows two polycarbonate sheets (thickness 180 μm) bonded by a sheet of pressure sensitive adhesive (thickness 86 μm) cut out to form a sample chamber. **c)** Plunging a sample of 20% DMSO into liquid nitrogen results in ice formation (top panel) and 30% DMSO forms a partial glass (middle panel). A solution of 40% DMSO forms a transparent amorphous glass, revealing the pattern of background surface, as assessed by visual inspection immediately after removing cassette from liquid nitrogen (bottom panel).

submersion in liquid nitrogen, where the sample remained for 30 min. For recovery from cryostorage, the sample was thawed in a 40˚C water bath and allowed to equilibrate in an excess of PBS to dilute intracellular CPA and relieve cryogenic stress. Viability post cryopreservation was assessed *in vitro* by excystation and PI exclusion assays. The heatmap shown in Fig 5B demonstrates the percent of successful cryopreservation outcomes stratified by oocyst age, CPA exposure time and temperature. The criterion for positive and negative cryopreservation outcome was determined based on the presence or absence of motile and full-bodied sporozoites excysted from thawed oocysts, as shown in the micrograph panel in Fig 5B. Since we often observe release of lysed sporozoites from thawed oocysts, we determined that quantification of excystation rate would lead to overestimation of viability, and therefore we prioritized assessment of sporozoite quality as a binary variable instead.

The heatmap in Fig 5B reveals variable responses to vitrification across oocyst age groups, which is likely reflective of their permeability to CPA. Permeabilization at 30˚C is an effective tool for delivery of sufficient CPA to support vitrification in oocysts older than 9 weeks, but not younger. As oocysts age, extended CPA exposure leads to a negative cryopreservation outcome, likely resulting from the accumulation of toxic concentrations of CPA related to increased oocyst permeability. This age-related effect is not apparent in oocysts permeabilized at 37˚C for 2 min during CPA exposure. Although no difference in viability after vitrification was noted across age groups by means of PI exclusion ($p = 0.59$, Fig 5C), we observed a reduced quality of excystation in 12-14-week-old oocysts in 15% of trials. This trend is consistent with our previous findings with *C. parvum* and suggests that aged *C. hominis* oocysts are

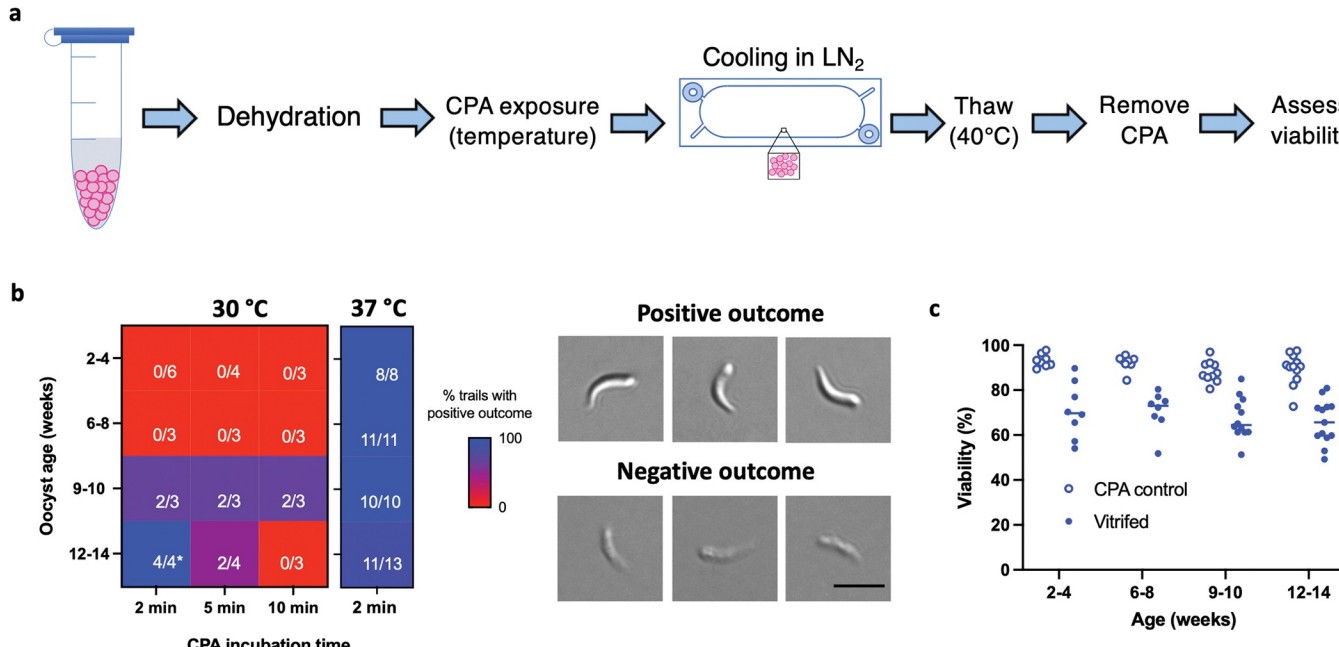

**Fig 5. *C. hominis* oocysts vitrified in cassettes are viable and excyst. a)** Prior to vitrification, oocysts were dehydrated in 1 M trehalose for 10 min and then incubated in 0.5 M trehalose/50% DMSO at either 30˚C or 37˚C for 2-, 5- or 10 min. Oocysts were then immediately loaded into cassettes and rapidly plunged into liquid nitrogen for 30 min. Oocysts were thawed by quickly transferring directly from liquid nitrogen to 40˚C water for 10 sec. After removal of CPA by incubation in excess PBS for 30 min, viability was assessed based on PI exclusion and fitness of excysted sporozoites. **b)** The heat map indicates the percent of positive cryopreservation outcomes in relation to oocyst age and time of incubation in DMSO at a corresponding permeabilization temperature. Cryopreservation outcome was determined by an excystation assay based on sporozoite morphology and motility. Positive outcome is defined by the presence of full-bodied and motile sporozoites of correct curvature, in contrast to thin and non-motile (presumed dead) sporozoites, as demonstrated in DIC micrographs (scale indicates 5 μm). Each box indicates the number of successful trials over the number of total trials attempted. The CPA protocol including 2 min incubation in 0.5 M trehalose/50% DMSO at 37˚C results in positive cryopreservation outcome irrespective of oocyst age. **c)** Oocyst viability was determined microscopically by means of PI exclusion, both before (CPA control) and after cryopreservation (vitrified) using the 2-min protocol of 0.5 M trehalose/50% DMSO exposure at 37˚C. No difference in after-cryopreservation viability was observed across age groups (One-way ANOVA; $p = 0.59$, F = 0.63, df = 3, Shapiro-Wilk normality test; $p > 0.16$ for all age groups, Brown–Forsythe homoscedasticity test; p = 0.67). Lines indicate mean (n = 8–13).

less likely to survive cryopreservation. Due to the consistent response of oocysts aged <10 weeks, the cryopreservation protocol including 2 min CPA exposure at 37˚C was prioritized for *in vivo* validation. This cryopreservation protocol allows recovery of 30% of oocysts, of which 70–75% are viable. Specimen loss is likely due to oocyst retention in the cassette (45.8± 8.7%) and lysis resulting from cryopreservation and thawing (22.7± 9.0%, S6 Fig).

## Infectivity of cryopreserved *C. hominis* in gnotobiotic piglet model

The infectivity of cryopreserved *C. hominis* oocysts was evaluated in newborn gnotobiotic piglets. Since the *in vitro* studies revealed that the fitness of oocysts for cryopreservation changes as a function of age, this effect was also evaluated *in vivo*. Piglets were infected with $10^5$ PI- oocysts from four different batches cryopreserved at either 7–9 weeks or 12–14 weeks of age. Figs 6 and S7 show that younger oocysts cooled in cassettes following 2 min CPA exposure at 37˚C were infectious to gnotobiotic piglets. Specifically, 80% of piglets (4/5) inoculated with cryopreserved oocysts developed a patent infection on 4-6th day post infection (dpi), with onset of shedding delayed by 1–3 days in comparison to control animals infected with fresh oocysts (Fig 6A). Histological examination of ileal sections however demonstrates that all piglets were infected (Fig 6B). In contrast, oocysts cryopreserved at an older age developed much delayed patent infection in 20% of piglets (1/5) (S8A Fig). Despite the lack of oocyst detection

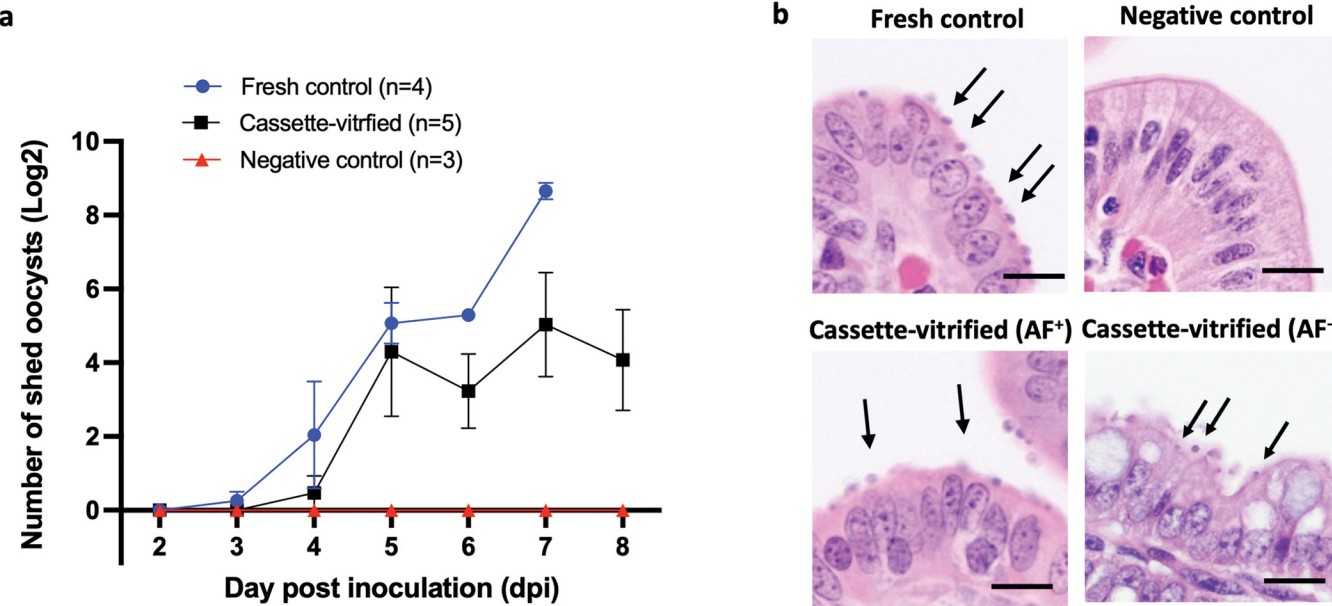

**Fig 6. *C. hominis* oocysts aged <10 weeks vitrified in cassettes are infectious to gnotobiotic piglets. a)** *C. hominis* oocysts aged 7–10 weeks originating from two different batches were cryopreserved in cassettes using the 2-min protocol of 0.5 M trehalose/50% DMSO exposure at 37°C. Gnotobiotic piglets were inoculated orally with 500,000 thawed PI⁻ oocysts (n = 5) in the presence of controls infected with 500,000 fresh, matched oocysts (n = 4) and negative controls (n = 3), of which 2 were uninoculated and 1 was inoculated with parasite frozen in absence of CPA. Fecal shedding of oocysts was determined daily by microscopic enumeration in 30 fields of acid-fast stained fecal smears examined under 1000x magnification. Eighty percent of piglets (4/5) produced a patent infection 1–3 days later than controls infected with fresh parasite. Values indicate mean of log transformed oocyst counts and bars indicate standard error. Untransformed infectivity data for each individual piglet can be found in S7 Fig. **b)** Hematoxylin and eosin-stained sections collected at 8 dpi from an experimental piglet show evidence of intestinal infection in both piglets infected with cryopreserved oocysts that tested negative and positive by acid fast (AF) staining. Arrows indicate intracellular parasite stages located at the apex of enterocytes. Scale indicates 20μm.

in feces, 80% of animals (4/5) became infected as evidenced by histological examination (S8B Fig). We expect that animals positive by histology but negative by fecal examination would have developed a patent infection if the observation period had been extended. Nevertheless, the observed delay in onset of fecal shedding and lower infectivity rate indicate that fewer oocysts survived cooling, suggesting that >12-week-old oocysts are less suitable for cryopreservation. This reduction in infectivity is analogous to that observed over time with unfrozen *C. hominis* oocysts, which loose infectivity after 24–28 weeks of storage at 4°C. Consistent with our *in vitro* findings summarized in the heatmap in Fig 5B, 13-week-old oocysts from a single batch cryopreserved after 2 min CPA exposure at 30°C infected 100% piglets (n = 3), even at a 10-fold reduced dose (S9 Fig). Although we advise cryopreservation of *C. hominis* at a younger age following CPA exposure at 37°C, a gentler permeabilization at 30°C could be considered for cryopreservation of oocysts older than 12 weeks. It is important to note that the handling procedure is sensitive to the surrounding temperature. For optimal results, all stages of the protocol should be performed at 21°C, with the exception of the brief thermal permeabilization step which is performed at higher temperatures. We observed that even small room temperature deviations (± 5°C) lead to lower viability after thawing (S10 Fig). Performance of the cassette vitrification protocol following 2 min CPA exposure at 37°C in age-matched *C. parvum* is comparable to *C. hominis* based on a preliminary *in vitro* assessment of viability and excystation (S11 Fig).

## Discussion

To our knowledge, this is the first report of successful cryopreservation of infectious *C. hominis* oocysts. To establish feasibility of vitrification for cryopreservation of *C. hominis*, microcapillaries were first tested due to their extreme cooling rates which enables vitrification even at low CPA concentrations [43,45,46]. Indeed, the microcapillary facilitated cryopreservation of infectious oocysts using the trehalose/DMSO cocktail in the apparent absence of intracellular DMSO following the protocol developed for *C. parvum*. Specifically, based on the insignificant cytotoxicity in the presence of high DMSO concentrations, it is inferred that oocysts do not uptake DMSO at ambient temperature (Figs 1B and S1A). However, oocyst dehydration was observed in the presence of high osmolarity CPA solutions, suggesting that *C. hominis* oocysts are vitrified in a shrunken state, as observed previously in *C. parvum* [39]. It is likely that oocyst dehydration increases the concentration of native sugars, which favors vitrification in the absence of exogenous intracellular CPAs. Consistent with this view is the fact that the genome of both *C. hominis* and *C. parvum* encode pathways for synthesis of amylopectin and trehalose [23,47], which are known for their glass forming potential [48,49]. Regardless of the precise mechanism that facilitates cryopreservation in the absence of intracellular DMSO, successful vitrification by ultra-rapid cooling in microcapillaries demonstrates that vitrification of oocysts is possible.

Recognizing the small volume of the microcapillary as a major limitation, the high aspect ratio cassette was designed for scale up of the sample volume to 100 μL. Due to its size and material properties, the thermal conductivity of the cassette is reduced in comparison to that of microcapillary, therefore intracellular CPA is necessary for vitrification. *C. hominis* oocyst permeabilization was achieved through a novel approach of thermal permeabilization. This approach was inspired by observations with *Mycobacterium* bacteria, which are also known for their impermeability [50]. A relationship between *M. chelonae* permeability and heat was suggested after observing increased fluidity of outermost lipids at temperatures ranging 30–60˚C [51]. The similarities in lipid composition and acid-fastness of mycobacterial cells and cryposporidial oocysts led us to consider thermal treatment as a method of permeabilization [41]. Oocysts suspended in the CPA cocktail consisting of trehalose/DMSO appear to uptake DMSO during thermal treatment, as evidenced by onset of morality due to DMSO toxicity. Specifically, trehalose-dehydrated *C. hominis* oocysts were exposed to DMSO under thermal permeabilization conditions. Although the concentration of DMSO accumulated inside oocysts is unknown, 2 min of 50% DMSO exposure at 37˚C resulted in successful cryopreservation. Additionally, permeabilization at 37˚C eliminated the variable response to CPA uptake related to oocyst age. It would be especially interesting to determine whether this permeabilization protocol can be also utilized to address between-species variability and streamline the development of a broadly applicable cryopreservation protocol for other *Cryptosporidium* species, specifically wild-type and transgenic lines of *C. parvum* and *C. tyzzeri*. Preliminary assessment of protocol translation to *C. parvum* indicates comparable performance based on measurement of viability and excystation, however infectivity of thawed oocysts remains to be determined in a mouse infection model in future studies. In comparison with the method previously published for cassette vitrification of *C. parvum* [40], chemical permeabilization of *C. parvum* by bleaching leads to slower intracellular uptake of CPA and therefore prolongs exposure to highly toxic concentrations of CPAs required for vitrification in cassettes. To alleviate accumulation of substantial toxicity in chemically permeabilized *C. parvum*, DMSO was delivered in two steps over 11 min of incubation [40]. Regardless of the permeabilization method applied, *C. parvum* and *C. hominis* vitrified using cassettes in a cocktail consisting of trehalose and 50% DMSO survive at a rate of 70–80% and 65–75%, respectively and are infectious *in vivo*.

In this work we redesigned major features of the vitrification cassette to improve its integrity, safety, and user experience. We recognize however that small differences in loading and cooling technique impact post-cryopreservation viability. The extended time of cassette loading may result in accumulation of CPA toxicity prior to cooling, whereas manual plunging of the device into liquid nitrogen is subject to variations and results in inconsistent cooling rates. These limitations can lead to reduced recovery of infectious parasites after thawing. For example, in this study, among five animals which developed intestinal infection after inoculation with cassette-vitrified oocysts, one did not show signs of patent infection on microscopic examination (Fig 6). It is expected that this animal would have excreted oocysts if the observation period had been extended, as histology confirmed infection. Given the correlation between infectious dose and the onset of patent infection, we suspect that variations in cassette handling led to recovery of lower proportion of infectious oocysts in this instance. Automation of loading and cryopreservation steps could potentially eliminate user-driven variations in cassette handling and increase the rate of successful cryopreservation outcomes.

Our previous work with *C. parvum* revealed the impact of oocyst age on outcome of cryopreservation by vitrification, both in microcapillaries and cassettes. Irrespective of permeabilization adjustments made to correct for parasite age, oocysts vitrified at the age >12 weeks showed evidence of reduced infectivity *in vitro* and *in vivo* [39,40]. The age-related decline in fitness for vitrification using cassettes was also shown to exist in *C. hominis*. It may be explained by the gradual depletion of intracellular glass-forming nutrients observed by others in aging oocysts [38]. Regardless of the mechanism by which *C. hominis* oocysts loose fitness for cryopreservation, the data described here suggest that vitrification using the cassette with 37°C permeabilization protocol should be reserved only for oocysts < 10 weeks to ensure optimal recovery of infectious product. Since oocysts become more permeable to CPA with age, a gentler permeabilization protocol at 30°C could potentially be utilized when cryopreserving oocysts older than 12 weeks. Although we have not systematically tested age-related response to cryopreservation by the microcapillary method, cryopreservation of *C. hominis* was found possible at the age of <4 weeks. It is unclear whether the observed age-related change in infectivity of cryopreserved inocula is reflective of the genetic bottleneck or the variability in oocyst permeability, water content and concentration of native biomolecules etc., leading to variable survivability. Future studies determining metabolomic, transcriptomic and genetic differences between fresh and cryopreserved oocysts across age groups can identify features that enhance survival.

While the stability of cryogenic storage in microcapillaries was demonstrated over 40 months, the cassette storage did not exceed 30 min. At the estimated cooling rate of ~$10^4$°C/min, the sample is expected to equilibrate to -196°C within one second. At this temperature the enzymatic activity stalls, therefore despite its brevity, a short-term cryogenic storage period is representative of a long-term cryostorage assuming specimens are not perturbed during storage [52]. Previously we demonstrated *C. parvum* infectivity remains the same after minutes- or months-long storage in liquid nitrogen [39].

We recognize several limitations in implementation of this method into routine laboratory practice. First, the method requires execution of a precise cooling and thawing procedure and continuous storage in the liquid phase of nitrogen to prevent formation of ice crystals. The handling technique during submersion in liquid nitrogen and thawing are critical determinants of vitrification outcome and therefore specific training may be required. Additionally, accidental exposure to liquid nitrogen vapors during cryostorage may result in ice crystallization. Future studies assessing the stability of glass in the cassettes stored in the vapor phase of nitrogen will inform the design of cryostorage solutions. Lastly, the capacity for amplification of oocysts recovered from cryostorage is restricted by the availability of animal models of infection.

The cryopreservation methods reported here for *C. hominis* oocysts will allow users to maintain this isolate without the need for continuous passage is animals. Propagation of *C. hominis* relies on timely executed surgical derivation of gnotobiotic piglets, which is vulnerable to unforeseen events such as natural disasters, pandemics and supply chain disruptions. Implementation of cryopreservation into routine laboratory practice will considerably reduce resource investment and animal sacrifice, as well as ensure continuity of this important isolate in the event of catastrophic loss of fresh specimens. Further, the procedure could potentially be used to support routine biobanking and dissemination of parasites. Broader access to specimens could increase progress in vaccine and drug discovery and facilitate an improved understanding of parasite biology. Importantly, cryogenic storage may also facilitate standardized testing of drug and vaccine candidates in clinical trials. This has been thus far hindered by batch-to-batch variability observed in parasite generated from different passages in gnotobiotic piglets [27]. The ability to biobank multiple aliquots from a single batch and thaw as needed can eliminate batch-to-batch variability and thus ascertain uniformity of the challenge dose. Additionally, the cassette volume is compatible for trial use, as it enables storage of multiple inocula in a single device. Lastly, inclusion of *C. hominis* in clinical trials is preferable to reflect its overwhelming contribution to disease burden. The ability to cryopreserve *C. hominis* makes it now possible to rigorously study cryptosporidiosis in the context of human infection.

## Materials and methods

### Ethics statement

All animal experiments were performed at Tufts University as accredited by the Association for Assessment and Accreditation of Laboratory Animal Care. All procedures involving swine species were conducted in compliance with study protocols G-2020-97 and G2020-116 approved by Tufts University Institutional Animal Care and Use Committee (IACUC) in accordance with the Guide for the Care and Use of Laboratory Animals (National Research Council).

### Oocyst source and collection method

*C. hominis*, TU502 isolate was originally derived from a Ugandan diarrheic child and has been maintained for over two decades at Tufts University by serial passage in gnotobiotic piglets every 3–4 months [22,53]. Oocysts were obtained from feces collected daily from infected animals. Following ether extraction of fats from fecal slurry, oocysts were purified using nycodenz density gradient as described previously [54] and were stored afterwards at 4˚C in PBS until use. *C. parvum*, Iowa isolate derived from infected calves was purchased from Bunch Grass Farms (Deary, ID) and was stored at 4˚C in PBS containing penicillin (100 U/mL) and streptomycin (100 μg/mL).

### Evaluation of oocyst dehydration in trehalose

The effects of trehalose on oocyst volume was examined as described in detail elsewhere [39]. Briefly, 2-week-old *C. hominis* oocysts were incubated in a solution of trehalose in PBS at concentrations ranging from 0.33–1 M for 10 min at room temperature. Oocyst volume ($V$) was regressed from the forward scatter signal recorded by flow cytometry (Accuri C6, BD Life Sciences) and compared to the starting volume ($V_0$) in PBS. The standard curve of volume to forward scatter signal was established using polystyrene spherical particles of different diameters, specifically 2, 3.4, 5.1, 7.4, 10.5 and 14.7 μm (Spherotech Inc., Lake Forest, IL). The relationship between particle diameter and forward scatter is defined by Pearson's correlation coefficient as

linear ($R^2$ = 0.93). The experiment was performed in triplicate and differences between treated and untreated oocysts were analyzed using one-way ANOVA.

## Permeabilization of oocysts to extracellular CPAs

Chemical and thermal permeabilization was employed throughout the study to facilitate intracellular uptake of DMSO in *C. hominis* oocysts. Comparison of chemical and thermal permeabilization between five batches of *C. hominis* was performed in 6-7-week-old oocysts (S1 Fig). ***a) Chemical permeabilization.*** Where indicated, oocysts were permeabilized chemically by bleach or alkane treatment. Bleaching solution was prepared by dilution of commercial bleach containing 8.25% sodium hypochlorite (Clorox Original, The Clorox Company, CA) in PBS. Alkane solution was prepared by dilution of a mixture composed of 90% d-limonene, 5% ethoxylated alcohol and 5% diethanolamine in water as described elsewhere [55]. For bleaching, 2- or 6-7-week-old *C. hominis* oocysts were incubated on ice either in 5–20% or 20% dilutions of bleach for 1 min, respectively, for younger and older oocysts (S1A and S1B Fig). For alkane permeabilization, 6-7-week-old *C. hominis* oocysts were vortexed with 5% dilution of the prepared alkane solution for 1 min at room temperature (S1B Fig). Following either permeabilization, oocysts were washed three times by suspension in PBS and centrifugation (18,000 ×g, 2 min). Loading of extracellular CPA occurred after chemical permeabilization. Specifically, oocysts were dehydrated in 1 M trehalose (10 min) followed by incubation with 30% (S1A Fig) or 50% DMSO (S1B Fig) for 30 min. (***b*) *Thermal permeabilization.*** Here, loading of CPA into *C. hominis* oocysts occurred during permeabilization by thermal treatment. Specifically, oocysts were incubated with 50% DMSO at 30˚C or 37˚C as described in detail below. For either permeabilization method, a control incubated with PBS for 30 min in lieu of DMSO was included for each species to normalize mortality. DMSO removal and measurement of DMSO-induced mortality in permeabilized oocysts are described in detail below. Experiments were performed in minimum of three replicates.

## Optimization of DMSO toxicity in thermally permeabilized C. hominis

The kinetics of cytotoxicity induced by DMSO were measured in thermally permeabilized *C. hominis* oocysts. Specifically, 2-12-week-old oocysts were first dehydrated in 1 M trehalose for 10 min and then incubated with 50% DMSO for 1–30 min at 30˚C or 37˚C. A control incubated with PBS for 5–30 min in lieu of DMSO was included for each temperature condition. Additionally, an unpermeabilized oocyst control incubated with DMSO at 21˚C was included as indicated (Fig 1B). DMSO was diluted out by incubation in excess PBS (1:100) for 30 min at room temperature and removed by centrifugation (18,000 ×g, 2 min). DMSO-induced mortality was measured by inclusion of propidium iodide using flow cytometry as described below and was normalized to a control incubated in PBS. Functional viability was measured by an excystation assay described in detail below. The experiment was performed in triplicate.

## Vitrification cassettes

The vitrification cassette consists of two sheets of polycarbonate laminate (thickness 180 μm) bonded by a sheet of pressure sensitive adhesive (thickness 86 μm) cut out to form a sample chamber. The top sheet of polycarbonate features two holes (diameter 2.5 mm) on opposite ends of the chamber covered with silicone tabs (diameter 8 mm, thickness 0.5 mm), one serving as a sample loading port and the other as a port for release of pressure during loading. The sample is loaded using a syringe affixed to a 28 ga. needle pierced at angle through the silicone tab, while pressure is released at the opposite port perforated with a needle (S5A Fig). The device is outfitted with an exit port, where the corner can be cut with scissors for sample

retrieval. The exit port is designed as a dead-end channel to prevent entry of biospecimen and to avoid splashes during cutting (S5B Fig). The cut cassette is then centrifuged in a 50 mL conical tube (200 ×g, 1 min) for easy removal of the contents from the device (S5C Fig). The cassette outer dimensions are 75 mm by 25 mm and the internal volume approximately 100 μL. Fabrication services were provided by Grace Bio-Labs (Bend, Oregon). The cassette can be produced in large quantities by Grace Bio-Labs by existing reel-to-reel manufacturing processes under catalog No. RD500893.

## Vitrification of C. hominis using microcapillaries

A detailed protocol for ultra-rapid vitrification and long term storage using microcapillaries is described elsewhere [39]. Here, 2,000,000 of <4-week-old *C. hominis* oocysts were treated with 5% bleach for 1 min as described above. Oocysts were pelleted by centrifugation (18,000 ×g, 2 min) and supernatant was removed. The pellet was first resuspended in 1 M trehalose and after 10 min incubation DMSO was added to achieve the final concentration of 0.5 M trehalose/30% DMSO in 20 μL volume. Immediately after addition of DMSO, the sample was loaded into 10 silica microcapillaries (Postnova Analytics Inc., catalog Z-FSS-200280, cut to 7 cm length) by capillary action and plunged into liquid nitrogen, one by one. The total time of capillary loading and immersion in liquid nitrogen did not exceed 5 min. Microcapillaries remained in liquid nitrogen for 10 min before thawing or were transferred to liquid nitrogen tank for long-term storage ranging from 2–40 months as indicated in S1 Table. Thawing was achieved by quickly transferring microcapillaries from liquid nitrogen to a 37°C water bath for 10 sec followed by immediate removal of biospecimen, one microcapillary at the time. The contents were expelled by flushing each microcapillary with 100 μL PBS using a syringe and a 30 ga. needle (BD). Expelled oocysts were incubated in a total of 1 mL PBS at room temperature for 30 min to facilitate CPA dilution. PBS was then removed by centrifugation (18,000 ×g, 2 min) and 100 μL of PBS was used to resuspend the pellet. Thawed oocysts were stored on ice prior to further testing. Oocysts used for animal infection were sterilized by incubation with 0.06% hexadecylpyridinium chloride monohydrate (HPC) for 10 min at room temperature and washed three times with PBS by centrifugation (18,000 ×g, 2 min). Sterilization with HPC on ice is strongly discouraged due to observed HPC precipitation and lysing effect on oocysts.

## Vitrification of C. hominis using cassette

An aliquot of 1,000,000–4,000,000 *C. hominis* oocysts was centrifuged (18,000 ×g, 2 min) and supernatant was removed. The pellet was resuspended in 50 μL of 1 M trehalose and incubated at room temperature for 10 min. Next, 50 μL of prewarmed 100% DMSO was added to dehydrated oocysts to achieve the concentration of 0.5 M trehalose/50% DMSO and was incubated on a heat block for either 2 min at 37°C or 2–10 min at 30°C. Immediately following incubation, 100 μL of sample was loaded into polycarbonate cassettes (Grace Bio-Labs, catalog No. RD500893) using syringe and a ≥ 28 ga. blunt needle. As sample is expelled through the loading port, displaced air is evacuated through the opposite port perforated by a needle as shown in S5A Fig. When the sample is deposited in the chamber, a syringe is used at the opposite port to remove remaining air through the needle to ensure uniform distribution of sample in the chamber. It is critical that loading is performed at temperatures not exceeding 21°C and is completed within 1 min. Immediately after loading, the cassette was plunged into liquid nitrogen. Although the sample reaches cryogenic temperatures nearly instantaneously at the cooling rate of ~$10^4$°C/min, oocysts were maintained in liquid nitrogen for 30 min prior to thawing. Oocysts exposed to cryoprotective cocktails for ~3 min (2 min incubation plus time equivalent to cassette loading ~40-50s) but not vitrified were also studied to assess toxic effects

of CPA in the absence of cryopreservation (CPA control). For thawing, the cassette was rapidly transferred from liquid nitrogen to a 40°C water bath for 10 seconds. It is critical that the cassette be rapidly transferred into liquid nitrogen for cooling, as well as from liquid nitrogen to the water bath during thawing, to avoid detrimental crystallization of ice. The sample was then recovered from cassettes by centrifugation. Specifically, exit channels were cut open on both ends with scissors and the cassette was placed in a 50 mL conical tube containing 3 mL of PBS for centrifugation (200 ×g, 1 min, S5C Fig). Expelled sample was further diluted with 2 mL PBS to reduce DMSO concentration to 1% and was incubated for 30 min at room temperature. The sample was then reduced to 100 μL by centrifugation (2,500 ×g, 10 min) and removal of supernatant. Thawed oocysts were stored on ice prior to further testing. Before animal infection, oocysts were suspended in in PBS containing penicillin (100 U/mL) and streptomycin (100 μg/mL).

## In vitro viability assays

Oocyst survival before and after vitrification was measured following removal of DMSO using viability and excystation assays. Viability was measured in whole oocysts by means of propidium iodide (PI) exclusion at 10 μg/mL using flow cytometry (Accuri C6, BD Life Sciences) or fluorescent microscopy (400x) as indicated in figure captions. For excystation assay, oocysts were incubated in 0.75% taurocholic acid in PBS at 37°C in a water bath for 30 min. In cryopreserved samples, the viability of excysted sporozoites was assessed qualitatively by microscopy on basis of shape, structure and observed motility (Fig 5B). Determination of the cryopreservation outcome based on microscopic assessment of excysted sporozoites was performed by non-blinded investigators. DIC micrographs (600x) of sporozoites shown in Figs 3C and 5B, were taken using Nikon Eclipse Ti-E microscope (Nikon Instruments Inc.). For measurement of CPA toxicity prior to vitrification, the functional viability of excysted sporozoites was evaluated quantitatively and compared to control oocysts incubated in identical conditions in PBS in lieu of CPA: trehalose and/or DMSO (Fig 2C). The rate of excystation was calculated as a percent of empty oocysts observed microscopically after incubation at 37°C either in taurocholic acid or PBS, for evaluation of induced or spontaneous excystation, respectively. Micrographs (400x) of sporozoites shown in Fig 2D were generated using an EVOS digital microscope (Life Technologies).

## Evaluation of infectivity in the gnotobiotic piglet model

The infectivity of cryopreserved *C. hominis* oocysts was tested in 1-2-day old gnotobiotic piglets (*Sus scrofa domesticus*). Healthy pregnant sows were obtained from Tufts Working Farm (North Grafton, MA) or from Parson's Farm (Hadley, MA). Piglets were surgically derived at Tufts University by Cesarean section performed in sterile surgical isolators by trained personnel under the supervision of a veterinary surgeon. Throughout the procedure sows were continuously anesthetized by inhalation with isoflurane (1 mL/4.5 kg) following sedation with tiletamine and zolazepam administered i.m. (Telazol, 800–1000 mg/animal). Following derivation and resuscitation, piglets were transferred sterilely into plastic microbiological isolator units and placed in 32°C room, where they were maintained under germ-free conditions throughout experimentation. Piglets were fed 3 times daily with a total of 500–700 mL/day of human infant milk replacer (Similac). Piglets were monitored a minimum of 3 times each day. After 24-48h of observation, healthy piglets were randomly distributed into groups of 1–4 piglets per isolator, such that experimental groups were kept separate from controls. To ensure that inocula were free of bacterial contamination, oocysts cryopreserved in microcapillaries were sterilized with 0.06% hexadecylpyridinium chloride monohydrate (Sigma) for 10 min at

room temperature, and fresh oocysts with 10% bleach (Clorox) for 10 min on ice, after which oocysts were washed three times with PBS by centrifugation (18,000 ×g, 2 min), while oocysts cryopreserved in cassettes were suspended in a solution containing penicillin and streptomycin before inoculation. Piglets were inoculated orally with 500,000–1,000,000 PI⁻ cryopreserved oocysts in PBS suspension. For each batch of experiments, a positive control group inoculated with 5,000–1,000,000 matched fresh oocysts was included and utilized for parasite propagation. Whenever litter size allowed, uninfected piglets were included as negative controls, which were either uninoculated or inoculated with matched dose of dead oocysts previously frozen in the absence of CPAs. Starting from 3 dpi piglets were monitored daily for fecal shedding of oocysts. A smear of fecal sample obtained by rectal swab was acid-fast stained and inspected microscopically for the presence of oocysts in 30 fields (1000x). At termination of experiment on 8 dpi at the latest, piglets were euthanized by intracardiac injection of phenytoin and pentobarbital (Beuthanasia or Somnasol, 100 mg/kg) following i.m. administration of ketamine (100 mg/kg) and xylazine (5 mg/kg). The sow was euthanized immediately following piglet derivation with intracardiac injection of Beuthanasia or Somnasol (100 mg/kg) while anesthetized by inhalation with isoflurane. Sections of small and large intestines were collected *postmortem* from piglets for preparation of histological sections stained with hematoxylin and eosin.

### 18S rRNA gene alignment

A portion of small subunit 18S ribosomal RNA (18S rRNA) gene from *C. hominis* oocysts recovered from piglets after 12 months of cryogenic storage in microcapillaries, was compared to a reference 18S rRNA gene of *C. hominis* (AF093491). Additionally, a sequence from matched *C. hominis* oocysts before cryopreservation and *C. parvum* (Iowa) oocysts were included as controls. Briefly, parasite DNA was obtained from fecal samples or from purified oocysts using QIAamp Fast DNA Stool Mini Kit (Qiagen). The 18S rRNA gene was then amplified in a nested PCR reaction using primers specific to *Cryptosporidium* 18S rRNA, specifically the set of primary primers: forward (5'-TTCTAGAGCTAATACATGCG-3') and reverse (5'-CCCATTTCCTTCGAAACAGGA-3') and secondary primers: forward (5'-GGA AGGGTTGTATTT ATTAGATAAAG-3') and reverse (5'-AAGGAGTAAGGAACAACC TCCA-3') [56], yielding a 780 bp long amplicon. Sanger sequencing services were provided by Genewiz (Cambridge, MA). Sequences were aligned with ClustalW software and the alignment displayed with BioEdit [57].

### Statistical analysis

All *in vitro* experiments were performed with a minimum of three replicates to ensure reproducibility. Graphing and statistical analyses of data were done using GraphPad Prism software (v 9.1.0, GraphPad Software, Inc.). All data included in ANOVA analyses were tested for requirement of normality with Shapiro-Wilk test and homoscedasticity with Brown-Forsythe test.

### Supporting information

**S1 Table. Summary of *C. hominis* infectivity in gnotobiotic piglet after cryopreservation using microcapillary method.**
(PDF)

**S1 Fig. Chemical permeabilization of *C. hominis* results in variable cryoprotective agent uptake.**
(PDF)

**S2 Fig.** *C. hominis* **oocysts vitrified in microcapillaries are infectious to gnotobiotic piglets (untransformed individual data plot).**
(PDF)

**S3 Fig. Alignment of 18s rRNA gene from** *C. hominis* **oocysts before and after cryopreservation in microcapillaries.**
(PDF)

**S4 Fig. High aspect ratio cassette design.**
(PDF)

**S5 Fig. Strategy for cassette loading and unloading.**
(PDF)

**S6 Fig. Recovery of oocyst sample from cassette.**
(PDF)

**S7 Fig.** *C. hominis* **oocysts aged $< 10$ weeks vitrified in cassettes are infectious to gnotobiotic piglets (untransformed individual data plot).**
(PDF)

**S8 Fig. Reduction in infectivity of** *C. hominis* **aged $> 12$ weeks cryopreserved in cassettes using 37˚C permeabilization protocol.**
(PDF)

**S9 Fig. Infectivity of** *C. hominis* **cryopreserved in cassettes using 30˚C permeabilization protocol.**
(PDF)

**S10 Fig. Cryopreservation protocol is sensitive to variation in room temperature.**
(PDF)

**S11 Fig. Comparison of the cryoprotective protocol between** *C. hominis* **and** *C. parvum* **oocysts.**
(PDF)

**S1 Protocol. Rapid cooling of** *C. hominis* **oocysts using high aspect ratio cassettes.**
(PDF)

**S1 Data. Supporting Information Data files.**
(ZIP)

## Acknowledgments

The authors acknowledge Jennipher Grudzien from Grace Bio-Labs for help with fabrication of the high-aspect ratio cassette.

## Author Contributions

**Conceptualization:** Justyna J. Jaskiewicz, Denise Ann E. Dayao, Derin Sevenler, Giovanni Widmer, Mehmet Toner, Saul Tzipori, Rebecca D. Sandlin.

**Data curation:** Justyna J. Jaskiewicz, Denise Ann E. Dayao, Donald Girouard, Derin Sevenler.

**Formal analysis:** Justyna J. Jaskiewicz, Denise Ann E. Dayao, Rebecca D. Sandlin.

**Funding acquisition:** Justyna J. Jaskiewicz, Giovanni Widmer, Mehmet Toner, Saul Tzipori, Rebecca D. Sandlin.

**Investigation:** Justyna J. Jaskiewicz, Denise Ann E. Dayao, Rebecca D. Sandlin.

**Methodology:** Justyna J. Jaskiewicz, Denise Ann E. Dayao, Donald Girouard, Derin Sevenler.

**Project administration:** Saul Tzipori, Rebecca D. Sandlin.

**Resources:** Saul Tzipori.

**Supervision:** Giovanni Widmer, Mehmet Toner, Saul Tzipori, Rebecca D. Sandlin.

**Validation:** Justyna J. Jaskiewicz, Denise Ann E. Dayao, Donald Girouard, Giovanni Widmer, Saul Tzipori.

**Visualization:** Justyna J. Jaskiewicz, Derin Sevenler.

**Writing – original draft:** Justyna J. Jaskiewicz.

**Writing – review & editing:** Justyna J. Jaskiewicz, Denise Ann E. Dayao, Derin Sevenler, Giovanni Widmer, Mehmet Toner, Saul Tzipori, Rebecca D. Sandlin.

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
