## [Decision Letter · Decision Letter 0]

29 Mar 2023

Dear Dr. Sandling,

Thank you very much for submitting your manuscript "Scalable cryopreservation of infectious Cryptosporidium hominis oocysts by vitrification." for consideration at PLOS Pathogens. As with all papers reviewed by the journal, your manuscript was reviewed by members of the editorial board and by several independent reviewers. The reviewers appreciated the attention to an important topic. Based on the reviews, we are likely to accept this manuscript for publication, providing that you modify the manuscript according to the review recommendations.

This is a technical paper focused on cryopreservation of C. hominis. Despite some advances, freezing parasites remains a major bottleneck of research with this important pathogen. Previous protocols have not found wide application yet due to several technical issues. All reviewers found the experimentation rigorous and supportive of the claims. While somewhat incremental in advance, this is an important step forward that should benefit multiple lines of Cryptosporidium research, most importantly, it should help to collect and store a variety of strains, both natural and transgenic. The editor agrees with the comments made by the reviewers that ask the authors to adjust some of the claims to the strength of the evidence, to discuss remaining limitations of the methods, and to step back a from the statement that C. hominis as the only appropriate model for research and drug and vaccine trials. Work over the last years e.g. has demonstrated C. parvum to be a solid predicted of drug susceptibility of C. hominis. Lastly, I might have overlooked this, but as this a pure methods paper, a detailed supplemental step by step protocol that includes where to purchase specialized materials like the cassettes would help to spread the technology throughout the field.

Sincerely,

Boris Striepen

Guest Editor

PLOS Pathogens

P'ng Loke

Section Editor

PLOS Pathogens

Kasturi Haldar

Editor-in-Chief

PLOS Pathogens

orcid.org/0000-0001-5065-158X

Michael Malim

Editor-in-Chief

PLOS Pathogens

orcid.org/0000-0002-7699-2064

This is a technical paper focused on cryopreservation of C. hominis. Despite some advances, freezing parasites remains a major bottleneck of research with this important pathogen. Previous protocol have not found wide application due to the special equipment needed. All reviewers found the experimentation rigorous and supportive of the claims. While somewhat incremental, this is an important step forward that should benefit multiple lines of Cryptosporidium research, most importantly, it should help to collect and store a variety of strains, both natural and transgenic. The editor agrees with the comments made by the reviewers that ask the authors to adjust some of the claims to the strength of the evidence, to discuss remaining limitations of the methods, and to step back a little from the statement that C. hominis as the only appropriate model for research and drug and vaccine trials. Work over the last years e.g. has demonstrated C. parvum to be a solid predicted of drug susceptibility of C. hominis. Lastly, I might have overlooked this, but as this a pure methods paper, a detailed supplemental step by step protocol that includes where to purchase specialized materials like the cassettes would help to spread the technology throughout the field.

Reviewer Comments (if any, and for reference):

Reviewer's Responses to Questions

**Part I - Summary**

Reviewer #1: This paper extends prior work from this group on cryopreservation of Cryptosporidium parvum to include Cryptosporidium hominis. The work appears to be technically excellent, and the manuscript is presented clearly. This would be an unusual paper for publication in PLoS Pathogens, but that is a decision for the editor, and there is no doubt this in an important technical advance for Cryptosporidium research. I have no major issues with the manuscript, but several editorial concerns noted below.

Reviewer #2: The authors present data about a new method for cryopreservation of C. hominis. This is a very welcome technology as there is currently no way to freeze down C. hominis and therefore this species must currently be maintained in gnobiotic piglet models. This animal model is challenging, expensive, and not easily accessed. Having a way to freeze down strains rather than regularly passage strains will be of great use.

Lines 84-85: I think a longer summary of the method that works for cryopreservation of C. Parvum is required. This includes the fact that you use a mix to trehalose and DMSO, that they are flash frozen in liquid nitrogen in microcapillaries (or the same cassette you use here for large scale freezing of C. Hominis), and stored in liquid phase. I don't think the precendent of this work is stated enough. It is important to demonstrate where the starting point was for C. Parvum and how that was used as a guide for the experiments detailed here. Currently this manuscript feels as if this method was invented de novo with little comparison to the c. Parvum method.

Also while it is a huge breakthrough to be able to cryopreserve C. Hominis, I think the authors overlook the challenges of this method. It requires liquid phase liquid nitrogen, access to the cassettes, large number of organisms, and training to be able to carry out this technique. I think it will be helpful to cryopreserve important strains, but will not solve all the current problems suggested in the text. For example, only those who can use the current animal model will be able to invest in this new method. I think a statement about the limitations of this approach should be included.

It seems that cryopreservation in microcapillaries will have more board appeal for individual Labotories to implement (doesn't require the cassettes and requires lower numbers of organisms). I think this method should be better highlighted in the manuscript. There will be several labs collecting specimens of C. hominis, and it would be much more feasible for them to cryopreserve their samples in smaller volumes.

Reviewer #3: There are many technical bottlenecks in research on cryptosporidiosis that is important in human and animal health, including the maintenance of Cryptosporidium species and isolates in the laboratory. Preservation of Cryptosporidium isolates in laboratory is a big burden as conventional cryopreservation protocols do not work. Among the two most common species used in research (C. parvum and C. hominis), C. parvum are usually propagated in calves both for materials and for keeping the isolates, or in knockout mice, whereas only one isolate of C. hominis could be maintained by multiple annual passages in gnotobiotic piglets at Tuft Univ. The procedures are not only expensive but requiring specific facilities. The problem become more of the issue in a laboratory that maintains multiple isolates including transgenic ones. Therefore, any progress in cryopreservation of Cryptosporidium oocysts is a significant progress in the field, including the progress described in this manuscript.

More specifically, while vitrification-based cryopreservation of C. parvum oocysts was reported several years ago (also from the same lab), that protocol is not applicable to C. hominis for which only one isolate (also serving as the reference strain) in the world is maintained exclusively in one laboratory. The authors renovated the C. parvum vitrification protocol and redesigned the cryo-container, making it possible to cryopreserve C. hominis oocysts for long-term maintenance of the parasite. Again, this is a significant progress in the field. It may also be applied to preserve oocysts newly isolated from the field before an appropriate propagation method is available.

**Part II – Major Issues: Key Experiments Required for Acceptance**

Reviewer #1: No Major issues

Reviewer #2: No experiments, but comments above should be addressed.

Reviewer #3: None.

**Part III – Minor Issues: Editorial and Data Presentation Modifications**

Reviewer #1: 1) Does the same approach work for Cryptosporidium parvum? If the authors have data validating this method with C parvum, it should be included in the supplement, along with any needed protocol modifications. I believe the potential importance of the method is greatest for simpler preservation of the growing number of transgenic C parvum strains than for single-lot production of C hominis that still can't be amplified by recipient labs.

2) I strongly disagree with the word "crucial" in line 330 regarding use of C hominis for clinical challenge trials. "Preferable" would be more accurate. Assuming comparable in vitro susceptibility of C hominis and C parvum, which should be measured for both species (ideally multiple isolates of both species), outcomes in an animal or a person will be dependent on drug exposure at the relevant site and not affected by the species. Studies to determine if the tissue distributions of C hominis and C parvum in people are the same or different may be warranted, but there is currently no reason to believe that outcomes in people would be affected by the species.

3) The data show clearly that cryopreserved Tu502 C hominis can be recovered by infecting piglets, but with a delay in the onset of shedding, so there is obviously a genetic bottleneck with freezing that may have important implications regarding the utility of the recovered parasites, with perhaps an exaggerated impact for older oocysts. The 16S sequencing in Fig S3 does little to address this. It is possible, for example, that a subset of the Tu502 parent strain has genetic or epigenetic features that enhance survival. This is worthy of mention in the discussion. Future studies to examine differences in the input and recovered populations might be very interesting, such as by whole genome sequencing to identify SNPs correlated with survival. Similarly, it would be very interesting to know if you progressively select for greater survival with sequential rounds of cryopreservation and recovery. Of course, such changes might be correlated with heartiness in other ways. This seems an issue worthy of discussion, as a limitation of the current method and also as a potential route for further investigation.

4) Line 44: Based on the reanalysis of GEMS data, it's no longer accurate to state cryptosporidiosis ranks 2nd amongst pathogens that cause life-threatening diarrhea. This should be corrected, and the GEMS reanalysis should be cited.

Reviewer #2: Abstract- the species used to achieve infection rate of 100% should be specified. Is this in vivo? In vitro? Which model? I think it is important to qualify this in the abstract.

Abstract- the abstract does not mention how the method is 'scalable'. I think this should be addressed in the abstract, or if it is not a significant part of the method it should be removed from the title.

Introduction line 64- Location should be added

line 68 should not be included. This manuscript is about the method of cryopresevation, not about the value of this specific isolate.

Last few sentences of the introduction are too brief. If you wan to summarise the results here, we need more detail (temperature? What is the composition of CPA?, etc).

Line 90- CPA should be spelled out, rather than abbreviated in the subsection title.

The only cryoprotective agent (CPA) used here is trehalose. The author should make this clear early on that they are only exploring trehalose. Or they should directly refer to trehalose in the text. By using the phrase CPA, it seems like several agents are being tested, while actually only trehalose is described in the manuscript.

Line. 131 and line 257- no evidence for this has been presented, so these comments should be removed.

In detailed protocol, final note detailing overall efficiency of freezing should be moved to the beginning of the protocol. This information is key to understanding how much material can be harvested from freezing, so should be highlighted accordingly.

Each results section could benefit from having multiple paragraphs to break up ideas.

How do you store the cassettes in the liquid nitrogen tank? This should be addressed.

Reviewer #3: The manuscript is also well-written. I have no major concerns, but a few minor ones for the authors to consider during the revision.

1. Is this newly developed protocol for C. hominis also applicable to cryopreservation of C. parvum oocysts? Could the authors give some discussion to compare this C. hominis protocol and the previous C. parvum protocol (some parameters and procedures)?

2. Lines 130 – 131 (bile salt): Please specify which one of the bile salts was used. The figure also lacks info on the bile salt.

3. Lines 225 – 228 (oocysts inoculated to piglets): Please briefly specify the number and batch info for convenience to the reader. Some other places in the Results section might also include brief M&M info, so readers do not have to frequently switch between Results and M&M sections.

4. Lines 228 – 236 (Negative in fecal examination but positive in histology of ileal sections): The investigators monitored oocyst shedding by microscopic examination of fecal smears (acid-fast stained), which might be less sensitive for low number of oocysts. PCR or qPCR might be more sensitive (not a requirement for new experiment, but worth trying if fecal samples are still available).

5. References: Most species names are not in italic fonts.

6. Fig. 4: Add unit (mm) to the diagram. Also specify that there are two sets of loading and exit ports (?). In panel (c), specify DMSO concentrations associated with the three forms (ice, ice/glass and glass). These figure annotation suggestions are mostly for easy reading and understanding, and applicable to some other figures.

PLOS authors have the option to publish the peer review history of their article (what does this mean?). If published, this will include your full peer review and any attached files.

Reviewer #1: No

Reviewer #2: No

Reviewer #3: **Yes: **Guan Zhu

Figure Files:

Data Requirements:

Reproducibility:

References:

---

## [Editor Report · Decision Letter 1]

15 May 2023

Dear Dr. Sandlin,

We are pleased to inform you that your manuscript 'Scalable cryopreservation of infectious Cryptosporidium hominis oocysts by vitrification.' has been provisionally accepted for publication in PLOS Pathogens.

Best regards,

Boris Striepen

Guest Editor

PLOS Pathogens

P'ng Loke

Section Editor

PLOS Pathogens

Kasturi Haldar

Editor-in-Chief

PLOS Pathogens

orcid.org/0000-0001-5065-158X

Michael Malim

Editor-in-Chief

PLOS Pathogens

orcid.org/0000-0002-7699-2064
---

## [Editor Report · Acceptance letter]

5 Jun 2023

Dear Dr. Sandlin,

We are delighted to inform you that your manuscript, "Scalable cryopreservation of infectious Cryptosporidium hominis oocysts by vitrification.," has been formally accepted for publication in PLOS Pathogens.

Best regards,

Kasturi Haldar

Editor-in-Chief

PLOS Pathogens

orcid.org/0000-0001-5065-158X

Michael Malim

Editor-in-Chief

PLOS Pathogens

orcid.org/0000-0002-7699-2064